# 'Unpacking' pathways to lymphoma and myeloma diagnosis: Do experiences align with the Model of Pathways to Treatment? Findings from a UK qualitative study with patients and relatives

Debra Howell [ID] ,[1] Ruth Hart,[1] Alexandra Smith,[1] Una Macleod,[2] Russell Patmore,[3] Eve Roman[1]

[1]Department of Health Sciences, University of York, York, UK
[2]Hull York Medical School, University of Hull, Hull, UK
[3]Queen's Centre for Oncology and Haematology, Castle Hill Hospital, Cottingham, Hull, UK

**Correspondence to**
Dr Debra Howell;
debra.howell@york.ac.uk

## ABSTRACT

**Objectives** To explore alignment of experiences before lymphoma and myeloma diagnosis with the appraisal, help seeking and diagnostic intervals in the Model of Pathways to Treatment (MPT).

**Design** A qualitative study using in-depth semistructured interviews with patients and relatives. Interviews were transcribed verbatim, anonymised and analysed using qualitative description.

**Setting** A UK population-based haematological malignancy patient cohort.

**Participants** Fifty-five patients (35 lymphoma, 20 myeloma: diagnosed 2014–2016) and 28 relatives participated, within around a year of the patient's diagnosis. Patients were selected from those in the cohort who had returned a questionnaire about their symptoms and help seeking, and consented to contact for further research. Sampling was purposive, to achieve maximum variation in age, sex and time to diagnosis.

**Results** Participants described time from symptom onset to diagnosis as ranging from several weeks to years. Pathways largely aligned with MPT components and help seeking could lead to the rapid investigations and identification of abnormalities. However, symptoms could be vague and/or inadvertently interpreted as other conditions, which if perpetuated, could cause diagnostic delay. The latter was associated with chaotic pathways, with activities rarely occurring only once or in a linear sequence. Rather, intermittent or ongoing processes were described, moving forward and backwards through intervals. This is 'unpacked' within five themes: (1) appraisal and reappraisal; (2) patient-initiated self-management/treatment; (3) initial help seeking; (4) re-presentation; and (5) patient-initiated actions, decisions and emotions during re-presentation. Within these themes, various healthcare professionals were consulted, often many times, as symptoms persisted/progressed. Input from family/friends was described as substantial, as was the extent to which information seeking occurred.

**Conclusion** Lymphoma and myeloma pathways align with the MPT, but do not fully capture the repetition and

## Strengths and limitations of this study

► In-depth interviews generated detailed data about complex experiences before diagnosis.
► The views of patients and relatives are captured, within contemporary healthcare systems.
► Maximum variation sampling ensured that findings are largely transferable to similar settings.
► Patients were largely interviewed within a year of diagnosis and some used diaries to promote recall.
► As the sample was derived from individuals returning a questionnaire about their symptoms and help seeking, the views of people unable to do this are not captured.

complexity described by participants. Time to diagnosis was often prolonged, despite the best efforts of patients, relatives and healthcare professionals. The impact of National Health Service England's Multi-diagnostic Disciplinary Centres on time to haematological cancer diagnosis remains to be seen.

## INTRODUCTION

Early diagnosis of symptomatic cancer can lead to improved quality of life and survival,[1 2] and has been a long-term UK policy priority, promoted via general practitioner (GP) referral guidelines and waiting-time targets.[3–5] The impact of such measures has been variable, however, with substantial reductions in time to diagnosis seen for some cancers, but limited benefits for others; and some actually seeing increases.[6] Further targets and initiatives introduced in the recent National Health Service (NHS) Long Term Plan,[7] as well as a broader range of potential pathways to diagnosis,[8 9] confirm this as an area that

continues to be clinically relevant, with early diagnosis still considered key to improving UK cancer survival.

Arising in blood and lymph forming tissues, haematological malignancies (leukaemia, lymphoma and myeloma) are collectively the fourth most common cancer among men (after prostate, lung and bowel) and women (after breast, lung and bowel) in economically developed regions of the world.[10–13] With diverse pathways and treatments, WHO currently recognises over 100 heterogeneous haematological malignancy subtypes, which display a range of behaviours and outcomes.[14] Myeloma, for example, generally presents in the bones and often requires intermittent treatment throughout a remitting–relapsing pathway, that is more typical of chronic disease than cancer. Non-Hodgkin lymphoma and Hodgkin lymphoma each have further subtypes, arise in various nodal/extranodal site(s) and may be indolent or aggressive. Myeloma and indolent lymphomas (eg, follicular lymphoma) are considered incurable at diagnosis, but can often be controlled for long periods of time. Aggressive lymphomas (eg, diffuse large B-cell lymphomas) are potentially curable with chemotherapy, but many patients still die from these diseases.

Patterns of disease onset and time to diagnosis of lymphoma and myeloma are varied. For example, patients may be asymptomatic and diagnosed incidentally or they may have multiple and/or severe symptoms, that can be diverse and similar to other self-limiting conditions (eg, fatigue, bone pain, swollen glands) to which they are often initially attributed or 'normalised'; symptoms may also be vague and non-specific in onset or intermittent.[15–17] Importantly, time to diagnosis can be prolonged and associated with multiple GP consultations before hospital referral,[18–23] myeloma being one of very few cancers that has seen increases in time to diagnosis in recent years, despite policy initiatives.[6] Emergency presentation is comparatively common in these cancers, a route linked to advanced disease, more complications and poorer survival than other pathways.[24–26] Conversely, fewer 2-week wait referrals are made to secondary care and more routine GP referrals than for other malignancies, suggesting cancer is not suspected at the time of hospital referral.

In the UK, a contemporary schematic template (known as the Model of Pathways to Treatment: MPT, figure 1)[27] is recommended for mapping and examining pathways to cancer diagnosis, along with a checklist to ensure consistent definitions, terminology, methods, data collection and reporting between studies.[28] The MPT comprises a series of intervals beginning with appraisal (patient appraisal of symptoms and self-management); followed by help seeking (decision to consult a healthcare professional (HCP), and make and attend an appointment); the diagnostic interval (HCP appraisal, investigation and referral); and the pretreatment interval (diagnosis to the start of treatment). Each interval is impacted by contributing factors, including patient actions, healthcare systems and disease characteristics. The MPT authors acknowledge that the intervals they describe may require 'unpacking' by cancer type.[27] The overarching aim of the current paper is to address this by unpacking the MPT, based on the experiences described by patients with lymphoma and myeloma and their relatives, in the time leading to diagnosis.

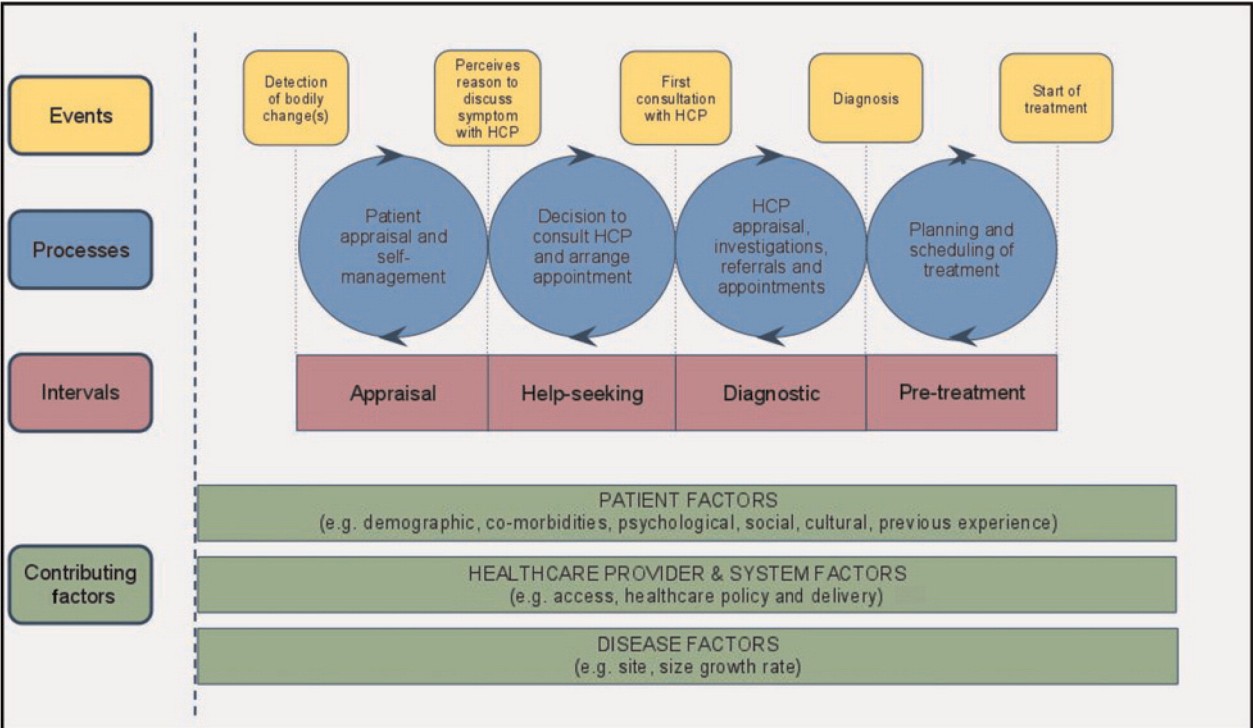

**Figure 1** Model of Pathways to Treatment (MPT).[27] HCP, healthcare professional.

## METHODS

Given the difficulties described above in achieving lymphoma and myeloma diagnosis, a mixed methods study was set up to explore the experiences of patients (and relatives) before diagnosis, which has, to date, resulted in several publications using data from medical records and qualitative interviews.[16 17 25] The current paper is based on the qualitative study phase, the methods of which are outlined below in accordance with the Consolidated criteria for Reporting Qualitative research.[29] Qualitative data are known to be well suited to investigating how people make sense of their situation, and how and why they make particular decisions.[30] In this study, our theoretical approach adhered to the principles of qualitative description,[31 32] an applied method facilitating the production of concrete, low inference descriptions with practical relevance. The MPT[27] was used as a framework from which to examine activities within the appraisal, help seeking and diagnostic intervals, as this is where the most significant delays are considered to occur.[17]

The qualitative study examining diagnostic pathways was based within the UK's Haematological Malignancy Research Network (HMRN: www.hmrn.org), detailed methods of which have been published elsewhere.[33 34] Briefly, HMRN is a specialist population-based cohort which was initiated in 2004. It tracks entire treatment pathways and collects data on all patients newly diagnosed with haematological malignancies across 14 hospitals (catchment population ~4 million) in the North of England (Yorkshire and Humberside). In addition to clinical data collection, HMRN patients are routinely asked to complete a questionnaire about their symptoms and help seeking.

Potential interviewees for the qualitative study were derived from patients completing the routine questionnaire in the HMRN area, who were aged ≥18 years, and within around 12 months of diagnosis. Sampling was purposive and designed to achieve maximum variation in age, sex and time to diagnosis. Ninety-two people were posted study material and invited to contact the research team if they wanted to participate; they were also invited to ask a relative/carer to contribute to the interview if they wished. Fifty-five patients (diagnosed 2014–2016; 20 with myeloma; 35 with lymphoma) and 28 relatives were interviewed, with data collection continuing until saturation. An overview of patient characteristics and the duration of intervals can be found in table 1, with further information provided in the .online supplementary table 1.

Interviews (conducted November 2015–May 2016 by RH and DH, both of whom were previously unknown to participants) were semistructured and guided by a schedule, developed from existing literature, as well as clinical advice from within the study team (box 1). With written consent, patients were asked to describe their pathways from first symptom(s) or health change(s) to diagnosis. Most interviews were carried out face to face in the patients home, although several were conducted in hospital or university settings, and a minority occurred by telephone. Interviews lasted around 45 min and were audio recorded, transcribed, anonymised and checked, with field notes used to confirm accuracy and support data analysis.

Analysis was iterative, running alongside and informing data collection (RH, DH). After review and familiarisation with transcripts, narrative summaries were produced from the data and pathways were mapped out visually and shared with the research team (ER, AS). Several rounds of coding were undertaken, focusing on different aspects of the data, including activities and events occurring within intervals defined in the MPT. Constant comparison (codes with codes, codes with data, codes with theory) and memoing[35] supported refinement of codes and identification of patterns and relationships. Codes and emerging themes were discussed by the team and

**Table 1** Overview of patient characteristics (n=55) and involvement of relatives by diagnostic group (n=28)

| Diagnosis | Patients (relatives*) | Total females | Total males | Median age years (range) | Duration of intervals (months/range) | | |
| --- | --- | --- | --- | --- | --- | --- | --- |
| | | | | | Appraisal/help seeking† | Diagnostic‡ | Total§ |
| **Non-Hodgkin lymphoma** | | | | | | | |
| Diffuse large B-cell | 12 (7) | 5 | 7 | 64 (48–81) | 0.5–13 | 1–15 | 2–17 |
| Follicular lymphoma | 9 (3) | 4 | 5 | 63 (39–84) | 0.5–12 | 1.5–12 | 2–15 |
| Marginal zone | 6 (1) | 4 | 2 | 62 (57–76) | 0.5–11 | 3–25 | 3.5–25.5 |
| Mantle cell | 3 (3) | 2 | 1 | 71 (70–75) | 1–6 | 2–10 | 2.5–10.5 |
| **Hodgkin lymphoma** | 5 (2) | 2 | 3 | 36 (23–56) | 0.5–12 | 1–12 | 3.5–13 |
| **Myeloma** | 20 (12) | 14 | 6 | 68 (43–78) | 0.5–7 | 0.5–17 | 2–18 |
| **Total** | **55 (28)** | **31** | **24** | **64 (23–84)** | **0.5–13** | **0.5–25** | **2–25.5** |

*Spouses, partners, children.

†Initial symptom/health change to first help seeking (from information estimated by patients completing the routine self-reported Haematological Malignancy Research Network (HMRN) questionnaire).

‡First help seeking to diagnosis (start point also estimated from the self-reported HMRN questionnaire).

§Initial symptom/health change to diagnosis.

**Box 1  Overview of interview schedule**

**Interviewees were asked to describe:**
1. Initial and subsequent symptoms and health changes
2. Interpretation of symptoms and health changes
3. Response to symptoms between onset and diagnosis: actions; activities; events
4. Help-seeking episode(s) and factors influencing this
5. Where, when and with whom help seeking occurred
6. Management by healthcare professionals (primary/secondary care)

refined until consensus was reached. Input from relatives was analysed alongside that of the patient, using the same methods.

## Patient and public involvement (PPI)

PPI is integral within HMRN, and takes place via a dedicated partnership, overseen by a lay committee. Patients from the partnership were involved in identifying the research questions underpinning the study during several focus group meetings, where they highlighted the difficulties they had experienced getting diagnosed; they were also involved in study design via our funding application.

**Theme 1: Appraisal and re-appraisal**
- Ongoing (intermittent or continuous), from symptom onset to diagnosis
- Became more precise and formal over time (use of diaries to document changes etc.)
- Other explanations for symptoms considered ('normalisation', self-limiting conditions)
- Information sought independently (internet, relatives, friends, colleagues etc.)
- Relatives were involved and adapted their behaviour to the patient's symptoms

**Theme 2: Patient initiated self management/treatment**
- Ongoing (intermittent or continuous), from symptom onset to diagnosis
- Coping mechanisms (distraction techniques, use of aids such as sticks, wheelchairs)
- Physical adaptation (stopping work, driving, hobbies)
- Lifestyle changes (healthy diet, exercise)
- Paying for treatments (over the counter medication, physiotherapy, osteopath)

**Theme 3: Initial help-seeking**
- All patients sought help from a GP or other HCP in primary care
- Impacted by advice from relatives, friends, colleagues etc.
- Broad health checks and general interventions requested
- Abnormality/cancer could be suspected and rapidly investigated by GP/HCP
- Other explanations for symptoms considered by GP/HCP (self-limiting conditions, menopause)

**Theme 4: Re-presentation**
- Common
- Could occur many times
- Response to:
  - worsening, progressive, new or alarming symptoms
  - inability to perform 'normal' activities
  - a growing feeling 'something' was wrong, but not knowing exactly what
  - lack of progress towards diagnosis
- Associated with negative investigations (blood tests, radiology, endoscopy, colonoscopy)
- Could be requested by the GP/HCP (safety netting)
- Dependent on initial advice/diagnosis given to patient by GP/HCP
- Occurred via various HCPs (dentist, physiotherapist, complementary therapist, NHS helpline)
- Could lead to use of emergency route (sent to A&E by GP/HCP, NHS helpline etc.)

**Theme 5: Patient initiated actions, decisions and emotions during re-presentation**
- Requesting more specific help (appraisal of specific sites, particular investigations)
- Presenting evidence to show symptom severity/persistence (diary, internet material)
- Giving GPs just the information the patient thought they would consider important
- Requesting re-presentation to the same GP, or a different/specifically identified GP
- Requesting hospital referral/admission
- Emotional outbursts and distress during consultations with GP/HCPs
- Involvement of relatives/others to advocate on behalf of the patient
- Use of emergency route (self-referral to A&E instigated by patient)
- Use of private healthcare to access specialist assessment

**Figure 2**  Summary of key themes and components. A&E, Accident and Emergency department; GP, general practitioner; HCP, Healthcare Professional; NHS, National Health Service.

Furthermore, patients routinely assist in the dissemination of all HMRN findings, which also occurs via our lay website: www.yhhn.org.

## FINDINGS

Participants described time from symptom onset to diagnosis as ranging from several weeks to numerous years. Within this disparate time frame, pathways largely aligned with the MPT, although experiences were more chaotic and did not always fit this model's distinct intervals or linear trajectory. This is 'unpacked' within five themes: (1) appraisal and reappraisal; (2) patient-initiated self-management/treatment; (3) initial help seeking; (4) re-presentation; and (5) patient-initiated actions, decisions and emotions during re-presentation. Each theme is described using illustrative verbatim quotes and summarised in figure 2.

### Appraisal and reappraisal

Patient appraisal of symptoms and health changes was described as ongoing by participants, with observation and monitoring occurring periodically or continuously from first symptom to diagnosis. When 'normal' health was not regained, appraisal was said to become more deliberate and rigorous. It involved, for example, checking and comparing affected and non-affected sites: 'My mind was going, "If you can feel (a lump)…both sides, then it's normal. If you can feel it on one side, then that's something to worry about". So I tried, I felt the other side, and again…' [Lymphoma patient 06: L06]. This was said to take place repeatedly over time: 'I couldn't decide whether my skin was a different colour…I'd look at it and think, "It looks strange", and then I'd look again, and it didn't' [L03].

Interviewees described reappraising persistent, progressive or new symptoms, which could be perceived to be related, or unrelated, to the initial problem. Unremitting symptoms sometimes led to more precise and formal appraisal, including documentation of health changes, pain, medication and weight: 'I kept a note, like week-by-week almost'…[L25]. A key element of appraisal involved patients and relatives considering numerous explanations for their symptoms over time. This included 'normalisation' of symptoms by attributing these to life-course issues (eg, age or menopause related) or events/ activities (eg, decorating), or self-limiting conditions: 'Could I have ME?' [L05]; 'Maybe it was an ulcer' [L26]; 'I was just thinking, "actually, is this becoming a mental health issue? Is it (the) first signs of dementia?"' [Relative of myeloma patient 20: M20].

Interviewees described information seeking from various sources during appraisal/reappraisal, particularly for ongoing symptoms; an issue not wholly acknowledged in the MPT. One myeloma patient consulted relatives about their family history (osteoporosis) as a potential explanation for symptoms. Others described questioning medically experienced friends, colleagues

or acquaintances: 'There's a doctor…lives further down the lane…I said [to him] "I don't really like asking but…"' [L03]. Many explored symptoms via the internet: 'I'd looked at all sorts of problems I thought (patient) might be having with bones"' [Relative, M20]. Internet searching was more effective when symptoms were typical of myeloma or lymphoma: 'I think I put in something like "itchy skin, night sweats, lump in neck"; Hodgkin's lymphoma came up, through a few different clicks' [L14]. This could lead to rapid help seeking/re-presentation: 'had I not seen (information), I'm not convinced I would have said anything to the doctor, but because I'd seen it and I thought, "Crying out loud…", I was like, "We've got to do something…someone needs to put all these pieces of the puzzle together"' [Relative, L14]. It could also lead to emergency presentation: 'the more I read (about lump), the more I was scared, and I just went to A&E…"' [L23]; or denial: 'Well, when I started reading about cancer I just switched it off, because I thought, "No, no, that's not right"' [L05].

Considerable involvement of relatives, friends and colleagues in appraisal/reappraisal was described, though not wholly depicted in the MPT: 'I just looked at his tummy and I thought, "that's an odd shape, that", and it just flashed through my mind and I thought "That could be a tumour"' [Relative: L20]. Some relatives said they adapted their own behaviour to accord with the patient, often over long time periods: 'patient's) height loss, was gradual, at first, so, I noticed…I stopped buying shoes with heels, because I was getting to be, you know, too tall (for him) when I was wearing shoes with heels…That had probably been going on…maybe a couple of years' [Relative: M18]; or they began to discuss the patient's symptoms and health changes between themselves: 'We knew there was something wrong…my daughter actually said, "You know, dad's not right at all, is he?"' [Relative: M02].

### Patient initiated self-management/treatment

Self-management was not solely associated with the appraisal interval as portrayed in the MPT, but was described as occurring throughout the pathway, particularly if patients or HCPs had normalised initial symptoms to self-limiting or life-course issues (eg, age, menopause): 'we're the kind of people who go to the doctor as a last resort, and we'd tried the obvious things ourselves, you know (lists remedies for indigestion), anything you think might ease it. But it didn't…' [Relative: L20].

Some patients said they used distraction techniques for relief, often long-term: 'I wake up because my tummy's hurting, and then I get up and start the day, and then I try and fill my day with things, so that I don't think about it…and, that was how it went and, we're probably talking a couple of years of that…just putting up with these different things' [L26]. Others reduced or stopped activities viewed to cause/aggravate problems: 'You'd stopped driving for about two months (before diagnosis), because you were in…pain' [Relative: M03]; [Interviewer: 'You gave up work 18 months ago…?'] 'Yeah…it were getting

a little bit difficult for me, and I didn't want to embarrass myself' [L16].

People described using aids over long time periods: 'I used to go to work with that belt (meant for weight-lifting), you know, on my back' [M15]. Others used sticks, wheelchairs or riser chairs to remain mobile. Lifestyle changes were made: 'I've had grumbling problems… probably at least two years…the kind…you treat yourself, with indigestion medicine, drinking water, trying to eat well, going for exercise, all those things that you think "This'll sort me out. I'm just getting a bit older and I need to look after myself" [L26]. Some took supplements 'we were having Complan drinks with whole milk…protein drinks, powders, anything to get something into his system…the weight back' [Relative: L20]; or self-medicated with analgesics, anti-inflammatories, antihistamines and gargles.

Long-term strategies were reported to manage debilitating tiredness, including napping or sleeping more: 'I were feeling tired all the time…I used to eat my sandwiches before my lunch…as I went round, and then as I stopped the van, I used to turn the ignition off, and lay on the seat, and go to sleep' [L16]; 'I tended to sleep a lot. I'd come home for lunch at 12 o'clock…fall asleep in the chair… almost every day…' [M12]. And then you'd come home and fall asleep, and you'd wake up and have tea, then fall asleep again, and then wake up for a couple of hours watching telly, and then go to bed' [Relative: M12].

Some people said they chose to pay for specific treatments, with mixed results: '(physiotherapy) made matters worse' [M18]; 'eventually, he were in that much pain, he couldn't go… (to physiotherapy)' [Relative: M15]; 'I went to an osteopath…which did help' [M12]. An 'herbalist' and an 'allergist' were also consulted and one person considered 'colonic irrigation'.

### Initial help seeking

Concurring with the MPT, all patients described help seeking for symptoms and attended their GP . The decision to seek help and the timing of this was often said to be influenced by relatives, friends or colleagues, who could encourage help seeking: 'My mum and friends (were) saying, "You…really ought to go to the doctors"' [M05]; or indeed discourage it: 'I had these lumps on my neck and I, I didn't take any notice of them…I said to a friend and she said…"Oh it's nothing, everybody gets lumps"' [L15].

A minority said initial help seeking led to the rapid identification of abnormality/suspected cancer, which was quickly investigated and diagnosed: 'The doctor… was lovely and she examined (patient)…and said "I can feel something, in there, and she booked him in…for a CT scan…endoscopy, yeah, and a blood test…on the scan they saw, there was a mass' [Relative: L20]; '(GP) did a very thorough check…he's always been very thorough… physical and, yeah, mental if you like…I've always found him absolutely superb. (He) listens not just to what you say, but what you don't say and, and sort of, reads the signals; [L01]; (GP) decided it was something serious,

and then you were sent for a scan…you had a biopsy, and it all happened very quickly' [Relative: L01].

Initial help seeking could include requests for broad health checks: 'I just felt, okay I'll go to the doctor's and… (say), "Well, I want an MOT…I seem a bit tired"…and it was a locum, and I explained my symptoms, and he sort of said, "Well, they seem a bit of an odd set of symptoms, so we'll give you…full blood tests", and thank goodness he did' [M05]; 'I did sort of ask for like a 'well woman' check, because I didn't in my mind feel like I was right' [L17]. General interventions were also sought: 'I was just really feeling fed up, because I was so tired…I went for a tonic, I said, "Can you give me just something to pick me up?"' [L34]. Alternatively, other conditions were initially suspected, particularly if these conditions had symptoms that were also associated with myeloma or lymphoma: 'I noticed a swelling at the top of my groin…(GP) said "that feels like a, a hernia to me…I'll sort you out an appointment to get you referred"' [L06].

## Re-presentation

As initial help seeking only rarely led to diagnosis, re-presentation was extremely common. This was said to occur quickly with increasingly alarming symptoms: '(I) went to the bank, and wet myself… and I thought, "Oh my God… what's going on here?" And I went (back) to the GP' [L09]; 'the sickness…he started bringing his evening meal up, and unable to sleep at night, the pressure on his tummy…he'd be walking the floor, you know, just generally, obviously something very wrong [L20]'. It could be delayed if the patient was told resolution may take time: I saw two (GPs)… [Interviewer: Who said?] 'Give it time' [M13]; or if symptoms were intermittent: 'he'd be ok for a while, and then, you know, maybe a few nights he'd be sick' [Relative: L20]; 'I carried on working, as you do…I didn't feel too bad really, some of the time' [L20].

Alternatively, repeat GP consultations could be delayed, particularly if initial/differential diagnoses were perpetuated: 'I noticed, like, a little pea-sized lump in my leg… my right calf…I thought… "Probably me varicose veins". I didn't rush, I went to see the doctor, regarding something else, and I just happened to mention to him about it… he had a feel of it, and he says "right, we'll send you off to have and ultrasound", you know, get it checked out. I asked (radiographer)…"Is it a varicose vein?" She says "yes". 'I got on with my life, I just carried on. I thought well fair enough, you know, its varicose veins…I never bothered when I started getting other lumps in my leg, I thought, "well it's all just part of it". (The following year) this lump was steadily growing…I started getting other lumps…the lumps joined together…starting to turn red and poke out of my skin…It just sort of exploded sort of thing, you know, it was frightening how fast it was growing…I went to see the doctor again' [L31].

Re-presentation often occurred on multiple occasions: 'I went back probably about three times between February and August… [M13]; 'all that time, just going backwards and forwards to the doctors' [Relative: L24]. Sometimes

this was associated with ongoing normal investigations: 'I just felt so tired. I'd just no energy, you know, and the daft part is, I think I'd three lots of blood tests done as well, to try and find out, you know, and yet none of 'em showed anything to do with the lymphoma' [L31]; 'The blood test revealed nothing, did it, and neither did the camera in the stomach' [L20]. Where abnormal blood tests were noted, rapid hospital referral was described: 'I'd had about four or five different blood tests, and it were all coming back negative' [M15]…'til these blood tests did come back abnormal, all of a sudden, so that's when they sent him up to haematology [Relative: M15]; '(GP) immediately, more or less said, I'm sending you into hospital. You're anaemic' [L24].

Repeat consultations sometimes occurred at the request of HCPs: '(GP) said to me if there are ever any changes with the lumps to inform him' [M12]. This was sometimes associated with HCPs expressing frustration about their inability to explain symptoms: '(GP) said: "come back in four weeks…I'm getting me books out…I don't understand what the symptoms are pointing me to" [L24, paraphrasing GP]. Interestingly, this patient's main concern ('night urination…combined with the loss of weight, that was what really got me, erm, worried'), remained unresolved after successful treatment, and at this point was considered unrelated, with urology review awaited.

Re-presentation also occurred due to a growing conviction something was really wrong: 'Normally I'm not somebody who goes to (the) doctors…it was just a feeling that something definitely wasn't right. Despite getting the first opinion' [L01]; My mobility was shocking by then…I was having great difficulty in doing anything, literally and I knew then that there was something…' [M03]; 'I weren't going to take "no" for an answer… because I knew there was something' [L34]. Doubt about an earlier explanation was often key: 'I was convinced there was something wrong with me, other than osteoporosis, because I thought "I shouldn't feel, this unwell, with osteoporosis."' [M14].

HCPs directed a number of patients towards emergency (re)presentation, either via A&E or a direct ward admission, often for worsening symptoms and to gain rapid access to healthcare. 'I saw the nurse, and she said, "Look, you know, go to A&E and let them have a look at it, so that's what we did' [L31]. Symptom complexity could result in consultation with multiple specialists even via this route, with the same patient being assessed by the A&E team, then a physiotherapist, vascular surgeon and plastic surgeon, before being told "it's a tumour" [L31].

## Patient-initiated actions, decisions and emotions during re-presentation

Some interviewees described requesting more specific input from their HCP at re-presentation, including direct appraisal: 'you need to see my neck again' [L14]; 'have a look while I'm standing up' [L34]; 'I said to (GP)…"Can you actually measure me?" which she did… she indicated… that I'd lost about two inches' [M14];

and particular investigations: 'I said, "Look, there's something wrong here. I need an X-ray"' [M13]; 'I said "I'm not picking up" and she agreed (to blood tests)' [L05].

Evidence (eg, diaries) was often produced by patients at re-presentation to demonstrate the extent and severity of symptoms: I thought, "(GP's) probably thinking I'm just a hypochondriac". But when I got the evidence of the weight that I was losing, I didn't feel that…(GP) could turn a blind eye to that.' [L25]. New information from re-appraisal (symptoms, mismatch between symptoms and preliminary diagnoses, and test results) or internet searching was also said to be passed on to HCPs: 'I said, "I think I've got lymphoma…I seem to have these lumps that haven't gone down at all…and I think now I'm getting, it's either anxiety, or I'm now getting B-symptoms, which is night sweats…"' [L06]. This could prompt action: 'I went back to the GP and said, "Well, (ENT) say it's TB or lymphoma", and the GP said, "Well, I don't think it's lymphoma, but, if they're saying TB, I'll send you for an X-ray"' [L07]; but not always: 'I'd been in pain for a long time, and I actually said, "I think I've got cancer" and (Dr) said to me, "You don't look like somebody who's got cancer"' [M14]; '(GP) said, "No, no, you haven't…lymphoma, old people get lymphoma. You're (under 40)…don't even think about that"' [L06]. Patients reported chasing-up information to give to HCPs (eg, results): 'I rang (surgery), three times a week, every week, to see if they'd had a reply' [Relative: L11].

Others blamed themselves for their delayed diagnosis, saying they only gave HCPs partial information about their symptoms: 'maybe I wasn't forthcoming with the information' [M03]; either purposely (only reporting symptoms they thought the HCP would considered important) or inadvertently: 'I didn't mention my general apathy, because I just thought it was just apathy to be honest' [M12]; 'many's the time I've come out thinking, "Oh damn, I didn't mention that"' [L24].

A few patients described feeling obliged to 'stick with' the first GP consulted: 'he knew the situation, and I kind of felt like, it'd have been going behind his back to go and see another GP' [L07]. Others did not: 'The very first doctor I saw. I saw him twice and then I knew after that I wasn't going to ask for that particular doctor' [M14]. Some reported re-presenting to HCPs they knew and trusted, were more experience, or just different: 'We would sometimes just ask for a different one, just to get a different opinion' [L34]. Decisions were also made to consult various other specialists: 'I'd started to suffer with mouth ulcers, and…I actually went to my dentist' [L30]; and advice was sought at unrelated hospital appointments: 'I went to see my Crohn's surgeon… my normal check-up, and I mentioned (the lump) to him, and he's the one really who pushed it forward fast…' [L32].

Lack of progress towards diagnosis could lead patients and relatives to ask their HCP to refer/admit them to hospital, which could be refused: '"Oh no, she doesn't need to go to hospital"' [Relative paraphrasing HCP, L35]. It also resulted in distress, anxiety and frustration:

"I was getting towards my wits' end, because I knew that there was something wrong, and I just felt as if (HCPs) weren't taking me seriously' [L06]. Occasionally, this resulted in emotional outbursts: 'The doctor said, "Have some antibiotics"… and we broke down and were just like, "This isn't right". So he called the hospital and got (patient) into hospital' [Relative, L14]; 'They thought it was anxiety, because I'd got myself in a state, and I was in a tearful mess at the doctor's, because of feeling ill and they're not getting anywhere…So she just thought it was an anxiety issue. It was like turned from one thing to something else' [L17].

Patients said their relatives often attended later GP appointments with them to advocate on their behalf: 'you become increasingly dependent on somebody to go with you…for reassurance and support' [M14]; 'We made and appointment…we had to insist (patient) needed to see someone, and we went together, and we said, "we need help here"' [L20]; '"I'm not waiting for this appointment", I said, "because (patient) can't wait". I said, "I want him to see (haematologist), as soon as!" I said, "I know you can do it, because I worked in the health-service" so she said, "Yes, okay". Otherwise, I honestly thought he could sit in the chair and just go (die). He was so ill' [Relative: L11].

Some patients and relatives recalled emergency presentation to hospital via A&E, which they initiated themselves: 'I was getting so desperate…I knew something was wrong, and I felt that nobody was really listening to what I was saying. I did actually go to A&E a couple of times. Well, once under our own steam, and once we rang 111, and they said "Go to A&E"' [M14]. Several people also mentioned eventually seeking private investigations and/or hospital referral, which they considered key to diagnosis. Access to private healthcare was said to be rapid: 'it bought me time' [L13]. Where GPs knew patients had private insurance, they were reported to suggest referral 'I went backwards and forwards to my GP, and I'm covered by my husband's private hospital thing, so (GP) suggested that I went to see a consultant back specialist' [L09]. More commonly, patients portrayed private care as an extreme but necessary measure: 'the timeframe for me was…it was too much, so I said "I want a private referral"' [M12].

## DISCUSSION
### Principal findings
Pathways to lymphoma and myeloma diagnosis largely aligned with the MPT.[27] However, further unpacking confirmed that trajectories were often much more complex and chaotic than depicted in the model, with activities rarely described as occurring on a single occasion or in a linear sequence. Rather, intermittent or ongoing processes were said to occur, that could move patients forward through intervals (eg, appraisal to help seeking), but also return them to the start of the pathway if there was no suspicion of cancer or a serious illness. In this context, repeat/continuous reappraisal could occur, with multiple re-presentations to HCPs and

associated anxiety and distress for patients. Symptoms were normalised[15] and self-management occurred across pathways, not only within the appraisal interval. Input from family and friends was substantial in all intervals, as was the extent to which independent information seeking occurred. Disease, patient and HCP factors (as presented in the MPT model) were found to influence each part of the pathway, with symptoms and events at one time-point (eg, GP reassurance about symptoms or a normal investigation) influencing later actions and time scales (eg, delayed patient re-presentation; delayed HCP referral to hospital). Similar experiences were reported by patients with myeloma and lymphoma (although more patients with myeloma were referred to hospital by the GP after an abnormal blood test); and various HCPs were consulted throughout the diagnostic interval.

### Strengths and weaknesses

This is the only qualitative study exploring pathways to diagnosis of lymphoma and myeloma in the context of the MPT, while taking an overview that tries to capture the interactions between people (patients, relatives and HCPs) and the events/activities taking place throughout the pathway. Our study included people diagnosed within contemporary healthcare systems, and after introduction of UK referral guidelines, targets and urgent diagnostic routes.[3–5] The sample is comparatively large (55 patients; 28 relatives) and in-depth discussions ensured that findings were based on personal experiences and perceptions. Inclusion of relatives is a unique feature and a particular strength, as these individuals (spouses, partners and adult children) were able to contribute their own views of the pathway and their involvement in events. Participation from relatives ranged from them taking a minor role (eg, reminding the patient about dates and events), to having significant input in describing the pathway and their own actions and emotions. Interviews were conducted within a year of diagnosis and the use of letters, calendars and diaries was encouraged to promote recall. Although generalisability is not an overarching aim of qualitative research, maximum sample variation increased the transferability[36] of findings to places with similar healthcare systems. Experiences may, however, differ in countries without universal healthcare provision, or where secondary/private healthcare gatekeepers are not GPs. Although we believe data saturation was reached, our sample was derived from individuals returning the routine HMRN questionnaire about symptoms and help seeking, so did not capture the views of people unable to do this, including those dying soon after diagnosis, who may have had more advanced stage disease. Furthermore, it is possible that symptoms attributed to myeloma or lymphoma by interviewees may in fact be unrelated to their cancer diagnosis.

### Comparison to other studies

We were unable to identify any other qualitative studies exploring the specific experiences of patients with myeloma and only two targeting lymphoma.[37 38] The latter were both UK based, and also reported descriptions from patients of challenging pathways and difficulties achieving diagnosis. Other quantitative work reports multiple GP appointments and delay before diagnosis (particularly among patients with myeloma); as well as the need to improve time to diagnosis.[18–23] Further studies have identified a group of cancers that are 'harder/difficult to suspect'.[18 23] Myeloma is among this group due to its propensity for non-specific symptoms commonly found in the general population (eg, back pain and fatigue).[19] Other 'harder to suspect' malignancies include pancreatic, stomach, lung, colon, brain and ovarian cancers, and Hodgkin lymphoma, and also typically have vague and intermittent symptoms, as well as complex pathways to diagnosis, including multiple GP consultations before hospital referral, more emergency presentations and longer time to diagnosis.[18 23 39–44] Conversely, malignancies with specific signs and symptoms (eg, breast cancer and breast lumps; melanoma and skin lesions), or affecting organs that can easily be examined (eg, testicular or thyroid cancer), generally have fewer GP consultations, more 2-week wait referrals for suspected cancer, fewer routine referrals and shorter time to diagnosis.[18 23 39 40 43] Given these differences, it is likely that the latter group will experience the linear compartmentalisation depicted in the MPT, whereas the former group is predisposed to the chaotic pathways described in this study.

### Meaning of the study and implications for practice

Our study clearly depicts the complexity of pathways to lymphoma and myeloma diagnosis. It clarifies the difficulties experienced in trying to identify the cause of what are often non-specific symptoms (intermittent, vague, slowly progressive, common in other illnesses). This difficulty could occur despite the best efforts of patients and relatives (eg, re-presenting to GPs for reassessment), as well as actions taken by GPs themselves, such as requesting investigations (eg, blood tests and imaging, and colonoscopies/endoscopies due to anaemia), which were often normal. Our findings highlight some of the reasons for delays in help seeking, re-presentation and diagnosis,[18–22] as well as potential explanations for the lack of urgent hospital referrals for suspected cancer and the high proportion of emergency presentations.[17 24–26] Our results also explain why interventions such as education campaigns targeting the (very broad) symptoms of these diseases would probably lack impact. Interestingly, some patients reported symptoms that were present before and after diagnosis (eg, nocturia: L24). While such symptoms were considered significant at the time, persistence after successful chemotherapy is likely to indicate that the issue was unrelated to their lymphoma, thus highlighting the difficulty of distinguishing malignant from benign disease. Moving forward, and in the absence of clear symptom signatures and blood tests[45–47] (although certain tests can rule myeloma in or out[48]), it is important that the complexity experienced by patients with myeloma and lymphoma, as

well as other 'harder to suspect' cancers, is acknowledged within strategies to reduce time to diagnosis, including those set out in the NHS Long Term Plan.[7] Importantly, this document introduced Rapid Diagnostic Centres (RDCs) that provide GPs with an alternative fast access pathway to secondary care for patients with symptoms that are worrying, but non-specific, so do not suggest referral to a specific clinical discipline. This could represent a significant change in the context of myeloma and lymphoma, as well as other 'harder to suspect' cancers, particularly if used in conjunction with rigorous safety netting in primary care,[4 49] of which some evidence was found in the current study. For members of the public, individual recognition of their own 'normal' is key, as well as having the confidence to pursue help seeking for unexplained symptoms.

### Unanswered questions and future research

Primary care HCPs have significant input into the care of patients later diagnosed with cancer, including those with emergency presentation.[50] To our knowledge, there have been no qualitative studies exploring GP experiences of managing patients with lymphoma or myeloma. Nor is there any work linking primary and secondary healthcare data, although such studies would clarify the occurrence and timing of events between symptom onset and diagnosis and promote understanding of these complex pathways. Exploring the impact of RDCs on time to diagnosis and disease stage at diagnosis will provide insights into the efficacy of this new pathway for patients with myeloma and lymphoma.

**Acknowledgements** The authors are grateful to the patients and relatives who took part in this study and shared their experiences at a difficult time.

**Contributors** DH, ER, AS, UM and RP devised the study, the methods and secured funding. RH and DH conducted interviews and analysed the data. UM and RP provided clinical input. DH prepared the draft manuscript. All authors edited and approved the final manuscript.

**Funding** This study was funded by Cancer Research UK (National Awareness and Early Diagnosis Initiative: C38860/A13509); Bloodwise (15037); and the National Institute for Health Research Programme Grant for Applied Research: RP-PG-0613-20002.

**Competing interests** None declared.

**Patient consent for publication** Not required.

**Ethics approval** REC 04/01205/69 (HMRN); REC 12/YH/0149 (qualitative study). The study was also registered with the UK Clinical Research Network Portfolio: UKCRN ID:11 963.

**Provenance and peer review** Not commissioned; externally peer reviewed.

**Data availability statement** No data are available. Ethical approvals do not permit data sharing. Patients did not consent to data sharing. The corresponding author can be contacted for further information.

**ORCID iD**

Debra Howell http://orcid.org/0000-0002-7521-7402

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
