## [Reviewer comments · BMJ Open]

ARTICLE DETAILS

TITLE (PROVISIONAL)	'Unpacking' pathways to lymphoma and myeloma diagnosis: Do experiences align with the Model of Pathways to Treatment? Findings from a UK qualitative study with patients and relatives
AUTHORS	Howell, Debra; Hart, Ruth; Smith, Alexandra; Macleod, Una; Patmore, Russell; Roman, Eve

VERSION 1 - REVIEW

REVIEWER	Katie Mills University of Cambridge, UK
REVIEW RETURNED	18-Oct-2019

GENERAL COMMENTS	Thank you for inviting me to review this paper. I believe it is an excellent addition to the current literature in this area. The inclusion of relatives to the interviews is novel and adds depth to the data collected, which is very much needed when describing such complex patient pathways to help-seeking. I have a few minor comments to the structure of the findings. Self-management is described as not solely associated with the appraisal interval, therefore it may be presented better as a stand alone theme which runs through both intervals. Could the initial section on self-management be linked with the later paragraph on patients seeking help from alternative therapies to create a new theme entitled "patient initiated management/treatment"? Theme 2 is rather long, with lots of very interesting data. I think it may be helpful for the reader to have sub-headings to highlight the messages described in each paragraph. I would also suggest moving the data describing referral to private medical care as a separate section as this does not seem to fit with the earlier part of this paragraph describing the influence of relatives and friends. If possible, it would be interesting to share in more detail the tools used by patients and relatives to promote recall and by how many. I would very happy to accept this for publication.
--

REVIEWER	Nicola Hall University of Sunderland, UK
REVIEW RETURNED	21-Oct-2019

GENERAL COMMENTS	This paper presents a well-written descriptive account of patient experiences of the pathways to diagnosis for lymphoma and myeloma within a UK NHS setting, with useful implications for policy/practice. It highlights and extends existing evidence about how experiences do not always follow the processes depicted in the MTP in a linear and compartmentalised way and the complexity of the diagnostic pathway for patients diagnosed with these cancers. The findings seem to extend those from a previously published paper (Howell et al, 2019) by including experiences of patients with myeloma (and their relatives). (It is mentioned on p 10, line 55 that this study (and one other) overlap with the present work, however it seems that this paper could potentially be based on some of the same lymphoma interviews from the previous Howell et al paper? If this is the case, this needs to be clarified.) The focus on aligning experiences with the Model of Pathways to treatment (MPT) is an important area to explore in terms of theoretical development, as well as adding to knowledge that can help with understanding and comparing the experiences of the pathways to diagnosis/treatment between patients diagnosed with different cancer-types and across different settings. In my opinion, I feel that the paper would benefit from consideration of the following issues before publication: Introduction:  • The study is focused on the time leading to diagnosis and aims to align the findings with the processes and intervals with the MPT. However, there seems to be a lack of rationale for the decision to only align experiences up to the point of diagnosis (ie why the focus was on the time leading to diagnosis only and the pre-treatment interval from the model was not included in the study/analysis?). Methods:  • The inclusion of relatives during the in-depth interviews is described as a strength of the study, however there is little explanation of how this added value, how this was accounted for in the analysis, or any description of the nature of the relationships to the patients. • Although the original questionnaire and potential participant pool was UK based, it is unclear where the qualitative sample was recruited from and whether this was from across the UK or restricted to one geographical area? Findings:  • The findings offer a very useful and clear description of the chaotic diagnostic pathways and the difficulties participants faced getting a diagnosis. I feel, however, that the decision behind the choice of “themes” may benefit from some additional clarification/explanation. For example, there could be a clearer rationale for why the MPT was used as a framework to “examine activities within the appraisal, help-seeking and diagnostic intervals”, yet only 2 “themes” are presented. The first theme aligns closely with the appraisal interval of the MPT and includes appraisal (and re-appraisal) and self-management, whilst the
---

second combines all the processes within the help-seeking and diagnostic intervals (including re-presentation). It is also acknowledged, however, that self-management occurred across the whole pathway, so it is unclear why, this was included only within the new “appraisal and re-appraisal” theme. The “help-seeking and representation to diagnosis” theme also seems to contain many important sub-themes that could possibly benefit from being highlighted (e.g. strategies used by patients to “convince” the HCP there was something wrong; signposting by HCPs etc...). This theme also overlaps with “appraisal and re-appraisal” in places. Although the content of the themes is well described, some re-consideration of how the findings section is “structured” may help to strengthen the rationale of why this new proposed thematic structure is felt to be more useful than the existing processes/intervals described in the MPT.

- Reported time from symptom onset to diagnosis was several weeks to numerous years. It would be useful to have a better idea from the patient identifiers about time to diagnosis to help contextualise the quotes (or this could be at least included within the participant characteristics table?) Gender and age would also be useful to include in the identifiers (if possible to include without being classed as identifiable).

- The patient identifiers distinguish between the quotes from myeloma and lymphoma patients, but it is unclear if any comparisons were made between the two patient groups in terms of their experiences/pathways to diagnosis. This is a shame particularly seeing as the rationale for the study was that the intervals of the MPT may require unpacking by cancer-type.

- Although there are a number of very important and well-described contributing factors included within the findings, these are not explicitly aligned to the 3 factor types within the MPT. It may be beneficial to describe this either in the findings or at least mention any similarities/differences within the discussion.

Discussion:

- I would have expected to see more discussion and evidence about how these findings compare with other studies that have also explored pathways to diagnosis/treatment in other conditions/cancers, as well as the possible implications in relation to theory development. For example, it is more than likely that conditions characterised by vague and common initial symptoms that are more difficult to diagnose will have more chaotic pathways and may result in more overlap in the processes described within the model than those who present with specific red flag symptoms? I feel that the comparison with a wider literature would strengthen this section and help to highlight the value of any novel findings. I would also recommend combining both strengths and weaknesses sections.

- Other limitations that should be mentioned include patients’ potential post-diagnostic attribution of general symptomatology to the diagnosis.

- The conclusion from the abstract mentions that diagnosis is often delayed despite the best efforts of patients, relatives and healthcare professionals. This is a point I feel could be extended or included within the discussion.

The following minor issues should also be addressed:

- Abstract: The final sentence in the conclusion needs correcting.
- Findings: Formatting of patient identifiers needs to be consistent
- p5 lines 11 and 37– grammatical / typo error to be corrected
- Discussion:

	 • p10 line 34 – the “interplay between intervals, people and events” may need some additional explanation? • p10 line 57/58 – it is unclear whether “this area” relates to research or practice? • Figure 2 - I don't think this figure adds anything to the paper and would recommend reconsideration of its inclusion.
--	--

VERSION 1 – AUTHOR RESPONSE

Reviewer(s)' Comments to Author:

Reviewer 1: Katie Mills, University of Cambridge, UK

Thank you for inviting me to review this paper. I believe it is an excellent addition to the current literature in this area. The inclusion of relatives to the interviews is novel and adds depth to the data collected, which is very much needed when describing such complex patient pathways to help-seeking. I would very happy to accept this for publication.

We are grateful for these positive comments.

I have a few minor comments to the structure of the findings.

- Self-management is described as not solely associated with the appraisal interval, therefore it may be presented better as a standalone theme which runs through both intervals. Could the initial section on self-management be linked with the later paragraph on patients seeking help from alternative therapies to create a new theme entitled "patient initiated management/treatment"?

Thank you for this useful comment. We have now made the changes suggested. Theme 3.2 is now Patient initiated self-management/treatment.

- Theme 2 is rather long, with lots of very interesting data. I think it may be helpful for the reader to have sub-headings to highlight the messages described in each paragraph.

With hindsight we agree and have separated the data into more distinct themes, which clarifies this section for the reader, as follows: 'Theme 3.3 Initial help-seeking'; 'Theme 3.4 Re-presentation'; and 'Theme 3.5 Patient-initiated actions, decisions and emotions during re-presentation'.

- I would also suggest moving the data describing referral to private medical care as a separate section as this does not seem to fit with the earlier part of this paragraph describing the influence of relatives and friends.

Again, we are grateful for this suggestion. We have now moved the section to the Theme 3.5: 'Patient initiated actions, decisions and emotions during representation'.

- If possible, it would be interesting to share in more detail the tools used by patients and relatives to promote recall and by how many.

Unfortunately, we are unable to provide this information, as it was not always routinely documented.

Reviewer 2: Nicola Hall, University of Sunderland, UK

This paper presents a well-written descriptive account of patient experiences of the pathways to diagnosis for lymphoma and myeloma within a UK NHS setting, with useful implications for policy/practice. It highlights and extends existing evidence about how experiences do not always follow the processes depicted in the MPT in a linear and compartmentalised way and the complexity of the diagnostic pathway for patients diagnosed with these cancers.

Thank you for these positive comments.

- The findings seem to extend those from a previously published paper (Howell et al, 2019) by including experiences of patients with myeloma (and their relatives). (It is mentioned on p 10, line 55 that this study (and one other) overlap with the present work, however it seems that this paper could potentially be based on some of the same lymphoma interviews from the previous Howell et al paper? If this is the case, this needs to be clarified.) The focus on aligning experiences with the Model of Pathways to treatment (MPT) is an important area to explore in terms of theoretical development, as well as adding to knowledge that can help with understanding and comparing the experiences of the pathways to diagnosis/treatment between patients diagnosed with different cancer-types and across different settings.

The reviewer is correct, this paper originates from the overarching study/interviews about diagnostic experiences, but focuses specifically on the MPT. We have made this clearer in the text at the beginning of the methods section, P4, as follows:

“Given the difficulties described above in achieving diagnosis of lymphoma and myeloma, a mixed methods study was set-up to explore the experiences of patients (and relatives) in the time leading to diagnosis, which has, to date, resulted in a several publications using data collected from medical records and qualitative interviews^{19,26,27}. The current paper is based on the qualitative section of the study, with methods outlined below in accordance with the consolidated criteria for reporting qualitative studies (COREQ)²⁸. Qualitative data are known to be well suited to investigating how people make sense of their situation, and how and why they make particular decisions²⁹. In this study, our theoretical approach adhered to the principles of qualitative description^{30,31}, an applied method facilitating the production of concrete, low inference descriptions with practical relevance. The MPT¹ was used as a framework from which to examine activities within the appraisal, help-seeking and diagnostic intervals.”

The paper would benefit from consideration of the following issues before publication:

Introduction:

The study is focused on the time leading to diagnosis and aims to align the findings with the processes and intervals with the MPT. However, there seems to be a lack of rationale for the decision to only align experiences up to the point of diagnosis (i.e. why the focus was on the time leading to diagnosis only and the pre-treatment interval from the model was not included in the study/analysis?).

The reviewer raises a valid point. The study was specifically set up to look at pathways to diagnosis and the factors associated with this. It focused on patient experiences before diagnosis, because studies have shown that this is when delays most likely occur. As the MPT is recommended for use to

ensure time-to-diagnosis research is comparable, we used this framework, but stopped at diagnosis. We have now clarified this in the introduction (P3, para 4), as follows.

“In the UK, a contemporary schematic template is recommended for mapping and examining pathways to cancer diagnosis, along with a checklist to ensure consistent definitions, terminology, methods, data collection and reporting between studies^{1,25}. The template is the Model of Pathways to Treatment (MPT: Figure 1)¹....”

Methods:

The inclusion of relatives during the in-depth interviews is described as a strength of the study, however there is little explanation of how this added value, how this was accounted for in the analysis, or any description of the nature of the relationships to the patients.

Strengths of including relatives are now captured within strengths and limitations bullets (p2):

- “The views of patients and relatives are captured, within contemporary healthcare systems.”

Inclusion of relatives is also discussed within the strengths and limitations section of the Discussion (p12, as follows:

“Inclusion of relatives is a unique feature and a particular strength as these individuals (most often spouses, but sometimes other family members), were able to contribute their own views of the pathway and describe their involvement in events. Participation in the interview ranged from taking a minor role (for example reminding the patient about dates and events), to having significant input in describing the pathway and their own actions.”

Regarding the analysis, the following section has been included at the end of the methods (p5):

“Input from relatives was analysed alongside that of the patient, using the same methods.”

Further details of patients who had relatives present are given in the new Supplementary Table S1.

Although the original questionnaire and potential participant pool was UK based, it is unclear where the qualitative sample was recruited from and whether this was from across the UK or restricted to one geographical area?

We apologise for this lack of clarity. We have added the following sections to the first and third paragraphs of the methods, p4, as follows:

“The qualitative study examining diagnostic pathways was based within the UK’s Haematological Malignancy Research Network (HMRN: www.hmrn.org), detailed methods of which have been published elsewhere^{16,17}. Briefly, HMRN is a specialist population-based cohort which was initiated in 2004. It tracks entire treatment pathways and collects data on all patients newly diagnosed with haematological malignancies across 14 hospitals (catchment population ~4 million) in the North of England (Yorkshire and Humberside). In addition to clinical data collection, HMRN patients are routinely asked to complete a questionnaire about their symptoms and help-seeking.”

Potential interviewees for the qualitative study were derived from patients completing the routine questionnaire in the HMRN area...”

Findings:

Findings offer a very useful and clear description of the chaotic diagnostic pathways and the difficulties participants faced getting a diagnosis. The decision behind the choice of “themes” may benefit from some additional clarification/ explanation. For example, there could be a clearer rationale for why the MPT was used as a framework to “examine activities within the appraisal, help-seeking and diagnostic intervals”, yet only 2 “themes” are presented. The first theme aligns closely with the appraisal interval of the MPT and includes appraisal (and re-appraisal) and self-management, whilst the second combines all the processes within the help-seeking and diagnostic intervals (including re-presentation). It is also acknowledged, however, that self-management occurred across the whole pathway, so it is unclear why, this was included only within the new “appraisal and re-appraisal” theme. The “help-seeking and representation to diagnosis” theme also seems to contain many important sub-themes that could possibly benefit from being highlighted (e.g. strategies used by patients to “convince” the HCP there was something wrong; signposting by HCPs etc...). This theme also overlaps with “appraisal and re-appraisal” in places. Although the content of the themes is well described, some re-consideration of how the findings section is “structured” may help to strengthen the rationale of why this new proposed thematic structure is felt to be more useful than the existing processes/intervals described in the MPT.

We thank the reviewer for this suggestion and have re-structured the themes accordingly (See ‘Results’ section and response to Reviewer 1’s comments).

Reported time from symptom onset to diagnosis was several weeks to numerous years. It would be useful to have a better idea from the patient identifiers about time to diagnosis to help contextualise the quotes (or this could be at least included within the participant characteristics table?) Gender and age would also be useful to include in the identifiers (if possible to include without being classed as identifiable).

This has now been included in Table 1, with further details in the Supplementary Table S1.

The patient identifiers distinguish between the quotes from myeloma and lymphoma patients, but it is unclear if any comparisons were made between the two patient groups in terms of their experiences/pathways to diagnosis. This is a shame particularly seeing as the rationale for the study was that the intervals of the MPT may require unpacking by cancer-type.

This was outside the scope of the current paper, although two additional qualitative manuscripts from the overarching study (referenced in the current paper’s methods, p4, para 1) were disease-specific and also alluded to the MPT, although the MPT was not the specific focus of the studies. Further details so the diagnoses/sub-types included are given in Supplementary Table S1.

Although there are a number of very important and well-described contributing factors included within the findings, these are not explicitly aligned to the 3 factor types within the MPT. It may be beneficial to describe this either in the findings or at least mention any similarities/differences within the discussion.

We were aware of this, but intentionally did not include contributing factors as specific categories as we felt it would dissect each theme too much. This was because patient, HCP and disease factors occurred within each theme and often overlapped. For example, vague symptoms (disease factor) impacted on both patient factors (delayed help-seeking) and HCP factors (delayed referral to secondary care). We have, however, commented more on this in the Principle findings section of the discussion (p12).

Discussion:

Expected to see more discussion and evidence about how these findings compare with other studies that have also explored pathways to diagnosis/treatment in other conditions/cancers, as well as the possible implications in relation to theory development. For example, it is more than likely that conditions characterised by vague and common initial symptoms that are more difficult to diagnose will have more chaotic pathways and may result in more overlap in the processes described within the model than those who present with specific red flag symptoms? Comparison with wider literature would strengthen this section and help to highlight the value of any novel findings.

Whilst there were no other qualitative studies with which to compare our findings, we have extended the section on how our findings compare to other quantitative studies, and the likelihood of these diseases aligning well with the MPT. We have also changed the heading to: 'Comparison to other studies', instead of 'Strengths and weaknesses compared to other studies' (p12). This section now reads:

“The latter were both UK-based, and also report patient descriptions of challenging pathways and difficulty achieving diagnosis. Other quantitative work continues to report multiple GP appointments and delay before diagnosis (particularly among myeloma patients); as well as the need to improve time-to-diagnosis²¹⁻²⁵. Further studies have identified a group of cancers that are ‘harder to suspect’^{21,37}. Myeloma is amongst this group due to its propensity for non-specific symptoms commonly found in the general population (e.g. back pain and fatigue²²). Other ‘harder to suspect’ malignancies include pancreatic, stomach, lung, colon, brain and ovarian cancers, and Hodgkin lymphoma, and also typically have vague symptoms and complex pathways to diagnosis, including multiple GP consultations before hospital referral, more emergency presentations and longer time-to-diagnosis^{21,37-42}. Conversely malignancies with specific signs and symptoms (e.g. breast cancer and breast lumps; melanoma and skin lesions), or relating to organs that can easily be examined (e.g. testicular or thyroid cancer), generally have a fewer GP consultations, more two-week wait referrals for suspected cancer, fewer routine referrals and shorter time-to-diagnosis^{21,37-39,42}. Given these differences, it is highly likely that the latter group will experience the linear compartmentalisation depicted in the MPT, whereas the former group are more pre-disposed to the chaotic pathways described in this report.”

I would also recommend combining both strengths and weaknesses sections.

The journal recommends that the discussion contains five paragraphs, using each of the headings presented in the originally paper. Consequently, we have changed the heading title slightly (see comment above), but have retained the five sections.

Other limitations that should be mentioned include patients' potential post-diagnostic attribution of general symptomatology to the diagnosis.

The following has been added to the paragraph on strengths and weaknesses, on p12:

“Furthermore, although this is likely to be rare (with hindsight), it is possible that symptoms attributed to myeloma or lymphoma by interviewees may in fact be unrelated to their cancer diagnosis.”

- The conclusion from the abstract mentions that diagnosis is often delayed despite the best efforts of patients, relatives and healthcare professionals. This is a point I feel could be extended or included within the discussion.

We have expanded what was written about this in the 'Meaning of the study' section of the 'Discussion' (p13), so that it now reads:

“Our study clearly depicts the complexity of pathways to lymphoma and myeloma diagnosis. It clarifies the difficulties experienced in trying to identify the cause of what are often non-specific symptoms (intermittent, vague, slowly progressive, common in other illnesses). This occurred often despite the best efforts of patients, relatives and HCPs, including patients re-presenting to their GPs and GPs requesting numerous investigations, which were often normal (including blood tests, imaging and colonoscopy/endoscopy, the latter often initiated due to anaemia.”

The following minor issues should also be addressed:

- Abstract: The final sentence in the conclusion needs correcting.

We have now amended the conclusion.

- Findings: Formatting of patient identifiers needs to be consistent

These are now consistent.

- Findings: p5 lines 11 and 37 – grammatical / typo error to be corrected

These have been corrected.

- Discussion: p10 line 34 – the “interplay between intervals, people and events” may need some additional explanation?

This has been re-phrased.

- Discussion: p10 line 57/58 – it is unclear whether “this area” relates to research or practice?
- This has been clarified.

Figure 2 – I don't think this figure adds anything to the paper and would recommend reconsideration of its inclusion.

Figure 2 has been re-formatted so it matches the themes more clearly. We feel this Figure provides an important summary and would prefer to retain it, unless the reviewer feels very strongly that it should be removed.

VERSION 2 – REVIEW

REVIEWER	Katie Mills University of Cambridge
REVIEW RETURNED	16-Dec-2019

GENERAL COMMENTS	Thank you to the authors for addressing each of my comments. I would be very happy to accept this paper for publication.
--

REVIEWER	Nicola Hall University of Sunderland, UK
REVIEW RETURNED	11-Jan-2020

GENERAL COMMENTS	Thank you for the opportunity to review the revised version of this interesting and well-written paper. All the comments have been addressed satisfactorily and I would be happy to recommend publication.
--